# Immune Deficiency in Jacobsen Syndrome: Molecular and Phenotypic Characterization

**DOI:** 10.3390/genes12081197

**Published:** 2021-07-31

**Authors:** Raquel Rodríguez-López, Fátima Gimeno-Ferrer, Elena Montesinos, Irene Ferrer-Bolufer, Carola Guzmán Luján, David Albuquerque, Carolina Monzó Cataluña, Virginia Ballesteros, Monserrat Aleu Pérez-Gramunt

**Affiliations:** 1Laboratory of Molecular Genetics, Clinical Analysis Service, Consorcio Hospital General de Valencia, 46014 Valencia, Spain; fatimafgf92@hotmail.com (F.G.-F.); ferrer_irebol@gva.es (I.F.-B.); guzman_reme@gva.es (C.G.L.); dav.albuquerque@gmail.com (D.A.); carolmonzoc@gmail.com (C.M.C.); 2Department of Pediatric, Consorcio Hospital General de Valencia, 46014 Valencia, Spain; montesinoselena@gmail.com (E.M.); ballesterosvir@gva.es (V.B.); maleupg@gmail.com (M.A.P.-G.)

**Keywords:** Jacobsen syndrome, 11q deletion, immunology, *FLI-1* gene

## Abstract

Jacobsen syndrome or JBS (OMIM #147791) is a contiguous gene syndrome caused by a deletion affecting the terminal q region of chromosome 11. The phenotype of patients with JBS is a specific syndromic phenotype predominately associated with hematological alterations. Complete and partial JBS are differentiated depending on which functional and causal genes are haploinsufficient in the patient. We describe the case of a 6-year-old Bulgarian boy in which it was possible to identify all of the major signs and symptoms listed by the Online Mendelian Inheritance in Man (OMIM) catalog using the Human Phenotype Ontology (HPO). Extensive blood and marrow tests revealed the existence of thrombocytopenia and leucopenia, specifically due to low levels of T and B cells and low levels of IgM. Genetic analysis using whole-genome single nucleotide polymorphisms (SNPs)/copy number variations (CNVs) microarray hybridization confirmed that the patient had the deletion arr[hg19]11q24.3q25(128,137,532–134,938,470)x1 in heterozygosis. This alteration was considered causal of partial JBS because the essential *BSX* and *NRGN* genes were not included, though 30 of the 96 HPO identifiers associated with this OMIM were identified in the patient. The deletion of the *FLI-1*, *ETS1*, *JAM3* and *THYN1* genes was considered to be directly associated with the immunodeficiency exhibited by the patient. Although immunodeficiency is widely accepted as a major sign of JBS, only constipation, bone marrow hypocellularity and recurrent respiratory infections have been included in the HPO as terms used to refer to the immunological defects in JBS. Exhaustive functional analysis and individual monitoring are required and should be mandatory for these patients.

## 1. Introduction:

Jacobsen syndrome or JBS (OMIM #147791), also known as 11q terminal deletion syndrome, is a rare genetic disorder caused by the loss of a continuous set of genes located on the long arm of chromosome 11. The deletion size varies from 2.9 to 20 Mb, and the breakpoints typically arise within sub-band 11q23.3, with deletions extending to the telomere [1]. A study by Mattina et al. found that approximately 85% of JBS cases were caused by de novo deletions, whilst the remaining 15% of cases were caused by imbalanced segregations and rearrangements of chromosomes [2]. The phenotype of complete JBS is considered when *BSX*, *NRGN*, *ETS-1*, *FLI-1* and *RICS* (*ARHGAP32*) genes are deleted [3]. However, when patients are haploinsufficient in some, but not all, of these genes, the patient is characterized as having partial JBS. 

This syndrome has a low incidence in the general population of approximately 1 in 100,000 births and a female-to-male ratio of 2:1 [4]. The phenotype of JBS is exhaustively detailed in the catalog of Human Phenotype Ontology (http://human-phenotype-ontology.github.io, accessed on 12 February 2021), and 96 HPO identifiers or terms describe the signs and symptoms. The most frequent clinical identifiers of JBS are dimorphic features, short stature, intellectual disability, congenital heart defects, thrombocytopenia, ophthalmologic problems, characteristic facial features and, for a minor population, urinary system malformations [2,5,6]. Despite the number of phenotypic characteristics detailed in the HPO, a deficient immune system has been identified in several reported cases of JBS, yet only constipation (HPO:0002019), bone marrow hypocellularity (HPO:0005528) and recurrent respiratory infections (HPO:0002205) are referred to in the HPO in relation to immunological defects. The molecular bases considered responsible for these characteristics have been attributed to the *FLI-1* gene deletion [7]. The description of the immunologic alterations already accepted as a component of the phenotype in a proportion of JBS patients suggests the existence of a syndromic primary immune deficiency that should be considered in the clinical evaluation of patients after diagnosis.

## 2. Case Report

A 6-year-old Bulgarian boy was referred to our clinical assistance laboratory for genetic consultation due to the presence of facial abnormalities and following an episode of loss of consciousness with generalized clonic movements. He was born at term following a normal pregnancy, to a non-consanguineous and healthy Bulgarian family. He had a birth weight of 2.280 kg, a length of 45 cm and a head circumference of 31 cm.

## 3. Methodology and Results

Phenotypic description: Phenotypic evaluation of the patient was performed using the HPO database and Face2Gene software (version: 6.0.3 (13)). The patient had abnormal facies, including a small chin, low-set ears with posterior positioning and anteversion, hypoplasia of the earlobes, hypoplasia of the lips, long philtrum, antimongoloid slant, mild inferior ectropion, broad sparse eyebrows, hypertelorism, ptosis, telecanthus, a prominent nasal bridge, flat occiput, short neck, pes planus, short stature, brief seizures and intrauterine growth retardation. Other characteristic signs and symptoms were thorax asymmetry with slight left protrusion, sacrococcygeal dimple, clinodactyly of the fifth fingers and thumb opposition. Regarding psychomotor development, he presented with intellectual disability, including cognitive impairment and attention deficit hyperactivity disorder, and diadochokinesia. His past medical history included diaphragmatic hernia surgery at the age of 9 months and a surgery for congenital cryptorchidism at the age of 3 years. He suffered from recurrent upper and lower respiratory infections (bronchitis and otitis) and constipation. His skin had eczematous-like lesions and hyperkeratosis without a barrier defect. 

Front and side photographs of the patient were uploaded to the Face2Gene software, and the most strongly suggested phenotype corresponded to Nijmegen breakage syndrome (NBS, OMIM 251260). Despite the molecular diagnosis of JBS, the facial analysis did not match the dysmorphology content for JBS in the linked databases (Figure 1). The published physical characteristics of patients affected by NBS are very similar to those of our patient.

Laboratory analysis: Hematology metrics, blood smear and flow cytometry analysis (FACS) were conducted. General biochemistry parameters, including glucose, iron and bone metabolism, were determined, and biochemical cardiovascular risk profile analysis was performed. Measurements of renal and liver function, blood ions levels, serum proteinogram and ceruloplasmin were carried out. Immunohematology Coombs test, immunoglobulins, complement and fecal elastase completed the blood analysis. Thrombocytopenia was detected, with platelet levels between 90 and 100 × 10^9^/L (reference values: 135–350 × 10^9^/L). The immune system analysis showed an immunodeficiency, with permanent and mild leukopenia; leukocyte levels were maintained between 1.5 and 3 × 10^9^/L (reference values: 4–12 × 10^9^/L), and lymphocyte measurements were between 0.6 and 1 × 10^9^/L (reference values: 1–5.5 × 10^9^/L). FACS analyses elucidated that T lymphocytes ranged between 600–750 cells/µL (reference values: 800–3500 cells/µL), with both CD4^+^ and CD8^+^ T cells levels lower than the established range (reference values: 300–2400 cells/µL and 300–1800 cells/µL, respectively). There was a similar finding for B lymphocytes, with levels of approximately 100 cells/µL (reference values: 200–2100 cells/µL). With respect to serum immunoglobulins levels, IgA and IgG levels were within the normal range, but IgM levels were between 40 and 45 mg/dL (reference values: 45–225 mg/dL).

Following the discovery of immunodeficiency, cytological study of the patient’s bone marrow and peripheral blood was undertaken to determine if the cytopenia had a peripheral or bone marrow origin. The cytological analysis found that myeloid, megakaryocytic and erythroid lineages were represented in a lower proportion. Mild–moderate myeloid and megakaryocytic hyperplasia indicated a peripheral origin for the patient’s cytopenia. There was no evidence of significant alteration in the morphology of the myeloid, megakaryocytic or erythroid cells, indicating possible dysmegakaryopoiesis. Peripheral blood analysis also revealed hypochromic microcytic anemia, despite markers of iron metabolism being within the normal range. Mild hyponatremia (around 130 mEq/L) was detected repeatedly. 

Image tests: To complete the phenotypic evaluation of the patient, magnetic resonance imaging (MRI) of the brain; diagnostic ultrasounds of the abdomen, thorax and heart; and an encephalogram were performed. The abdominal ultrasound revealed cryptorchidism, with undescended testes present in the inguinal canal, which were of different sizes. Following the identification of the position and size of the testes, the defect was straightened out to the correct anatomic position.

The echocardiogram revealed that the patient suffered an intermittent right bundle branch block. Echocardiography showed levocardia (HPO: 4383) and levoapex (HPO: 1629), as well as a permeable foramen ovale of 2–3 mm and a bicuspid aortic valve (HPO: 1646). Despite these congenital heart malformations, cardiological function was normal. 

Genetic tests: Analysis using high-resolution whole-genome SNPs/CNVs microarray hybridization was performed using Cytogenetics CytoScan HD Array (Affymetrix Inc., Santa Clara, CA, USA). More than 3 million markers covering the entire genome were analyzed according to the manufacturer’s instructions. The results evidenced a heterozygous loss of 6.8 Mb in the long arm of chromosome 11: the deletion arr[hg19]11q24.3q25(128,137,532–134,938,470)x1 (Figure 2). Partial JBS was diagnosed due to haploinsufficiency of three of the five genes considered as essential for JBS development: *ETS-1*, *FLI-1* and *RICS* (*ARHGAP32*). The deleted region encompasses 71 genes, of which 22 are OMIM genes (Table 1). This was determined to be a de novo deletion because both parents had normal karyotypes, as assessed by high-resolution G-banding karyotype analysis.

The rest of the detected CNVs were considered normal due to their existence in the reference population database, which suggests that they do not have any relation to the phenotype or deleterious capability. Loss of heterozygosity regions did not reveal consanguinity or uniparental disomy. Germline mosaicism was also ruled out as a cause.

## 4. Discussion

Since the description of JBS in 1973, over 200 cases have been reported; all of them have been characterized by deletions affecting the telomeric q region of chromosome 11. This terminal haploinsufficiency in chromosome 11 could affect more than 100 different genes. The diagnosis of complete JBS is established when the genes *BSX*, *NRGN*, *ETS-1*, *FLI-1* and *RICS* (*ARHGAP32*) are included in the deletion [3]. In particular, deletion within the 11q24.1 band appears to be critical for the expression of the JBS phenotype. JBS can have a wide variety of signs and symptoms, and these identifiers are exhaustively listed in the HPO (http://human-phenotype-ontology.github.io, accessed on 12 February 2021). The relationship between combinations of deleted genes and distinct sets of characteristics is being investigated in patients with a molecular diagnosis of partial JBS to determine correlations between specific genetic alterations and phenotypes of JBS. The existence of 30 out of 96 HPO terms or identifiers in our case was more concordant with the clinical phenotype described for complete JBS, although the underlying causal genetic alteration corresponded to partial JBS.

Face2Gene is an increasingly used tool for phenotyping and is considered to be a standard of professional excellence in genetics services due to its proven reliability. However, in our case, Face2Gene did not correlate the patient’s appearance with facial schemes associated with Jacobsen syndrome. The software instead suggested a differential diagnosis of NBS, which we considered to be a plausible suggestion, and had we not had a molecular diagnosis of JBS, we would have investigated the susceptibility gene for NBS (NBS1).

Reports of JBS cases in the literature have found quite distinct cytogenetic alterations, making the diagnosis of JBS more difficult. Gadzickit and colleagues described a case of JBS caused by an inversion: inv(11)(p15q24) [12]. Other cases of JBS presented in the literature have been due to translocations between chromosome 11 and other chromosomes, either autosomal [13,14,15,16,17] or sexual [18], that cause a partial monosomy of the 11q arm. Cases with the typical mosaic deletion of JBS have also been described [19]. The partial deletion arr[hg19]11q24.3q25(128,137,532–134,938,470)x1 found in our patient contained 71 genes that overlapped with the most common cytogenetic causes for JBS.

The phenotype of JBS patients specifically includes defects in immune system function. Some authors have suggested that JBS should be considered primarily as an immunodeficiency syndrome [7]. Early recognition and diagnosis of JBS patients lead to earlier therapeutic intervention, which results in the prevention of recurrent infections and subsequent organ damage.

Occasionally, authors have related the observed immunodeficiency seen in JBS to Paris-Trousseau syndrome (PTS, OMIM #188025). PTS is characterized by thrombocytopenia with abnormal platelets with alpha-granules and dysmegakaryopoiesis and is caused by deletions in the 11q23 region, which is located centromeric to the susceptibility region for JBS [20,21]. Due to this close proximity, some authors have suggested that PTS would be a variant of JBS when the deletion overlaps both 11q23 and 11q24 bands [7,21]. JBS patients normally carry deletions that include the proto-oncogene Friend leukemia virus integration 1 (*FLI-1*) [7]. This gene encodes an erythroblast transformation-specific (ETS) transcription factor that activates the expression of TGFBR2. The heterozygous deletion of *FL1-1* has been proposed to cause functional impairment and deficiency of T helper cells, dysmegakaryopoiesis and low serum level of IgM in patients [9,19,22], as well as in murine models [23]. Therefore, *FL1-1* is considered to be the most relevant gene for inducing thrombocytopenia, and deletion of *FL1-1* has also been associated with high levels of megakaryocytes and T and B leukocytes and low serum levels of IgM. However, *FLI-1* is not the only candidate gene for causing immunological defects in JBS and PTS patients. *ETS1*, *NFRKB*, *JAM3* and *THYN1* genes could also play a role in the presence of immunodeficiency. ETS1 encodes a transcription factor from the ETS family, which has been proposed to be involved in hematopoiesis, thrombocytopenia, pancytopenia and even in the formation of micromegakaryocytes. NFRKB encodes a DNA-binding protein (R-κB) that regulates *IL2R* α-chain gene expression, which is crucial for the activation of T cells. *JAM3* is a member of the junction adhesion molecule family located in the cell surface, with a high expression in CD8^+^ T cells, as well as activated T lymphocytes. Additionally, *JAM3* is highly expressed on megakaryocytes and platelets and has been proposed to participate in the inflammatory process mediated by monocytes. Finally, the *THYN1* gene encodes a thymocyte nuclear protein that has been suggested to be involved in apoptosis and is present in CD34+ hematopoietic stem and progenitor cells. The expression of the *THYN1* gene in these immune cells leads us to propose its implication in the immunodeficiency of JBS patients, along with *FLI-1*, *ETS1*, *NFRKB* and *JAM3*.

The first case of JBS with immunologic deficiency was reported in 1998 [24]. The patient was a 12-year-old girl carrying a deletion in the candidate chromosome region and suffering from moderate thrombocytopenia and antecedents of upper respiratory infections with decreased levels of IgA and IgM. Other similar cases have also been described [19,25,26]. Recently, the association of severe thrombocytopenia with JBS has become more widely acknowledged [19,22].

The dysmorphic facies seen in patients with JBS have been correlated with the deletion of the *BARX2* gene because of its role in the formation of the ectodermal lining of the mandibular and maxillonasal tissues and the tissues surrounding the eye during craniofacial development. 

The HPO database only includes eczema (HPO: 964) among the features of JBS that relate to skin and hair. The gene *ST14* was also included in the deletion in our patient. Per OMIM, ST14 is related to ichthyosis with hypotrichosis syndrome in a recessive inheritance manner. Thinning hair and hyperkeratosis were evident in our patient, suggesting an alteration associated with the structure of the stratum corneum.

The congenital heart defects described in some of the reported cases in the literature were also observed in our patient. The 7 Mb terminal region of the 11q chromosome has been proposed as a critical region for a putative causative gene or genes for human congenital heart disease, such as hypoplastic left heart associated with severe thrombocytopenia [5,12,27]. The same authors also proposed that ETS1 haploinsufficiency is the genetic cause of this phenotype, due to its expression in the endocardium and neural crest during cardiac development in murine models [28]. 

The mild hyponatremia detected recurrently in our patient will be controlled, but the disequilibrium of ions was not considered until the last medical follow-up.

## 5. Conclusions

Following an exhaustive review of the signs and symptoms of partial and complete JBS phenotypes, we suggest a revision of the described phenotype and the inclusion of additional identifiers. Primary immune deficiency should be included as a major criterion of diagnosis, with a specific description of the functional effect on the immune system. Immunological analysis should be performed in each patient diagnosed with JBS, since early recognition of immunodeficiency enables earlier therapeutic intervention and prevention of recurrent infections.

For genotype–phenotype correlation, phenotype description with the HPO database and molecular characterization of the genetic alteration in each case are essential. A detailed clinical description and the differentiation between Jacobsen and Paris-Trousseau syndromes should determine if they are two variants of the same syndrome or different syndromes. 

The failure of the Face2Gene software in predicting the Jacobsen syndrome phenotype based on photographs of the patient suggests high variability in facial features of patients with JBS.

## Figures and Tables

**Figure 1 genes-12-01197-f001:**
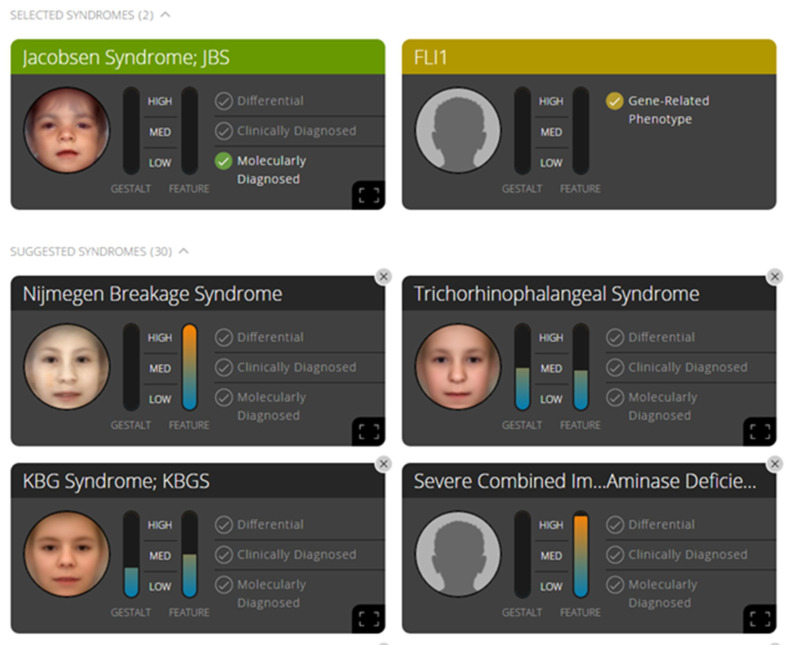
Prediction of the Face2Gene software, after front and side photographs of the patient were uploaded, was Nijmegen breakage syndrome (NBS, OMIM 251260) phenotype. Even though the molecular diagnosis of JBS was included, the facial analysis did not match with JBS.

**Figure 2 genes-12-01197-f002:**
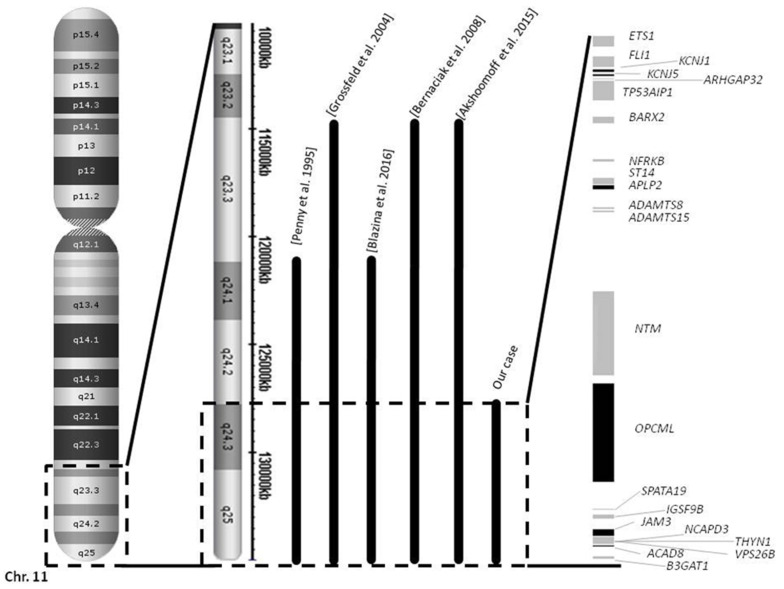
Deletion of 11q24.3-q25 carried by a partial JBS patient. Chromosome 11, showing the critical region of JBS, is shown on the left. Comparison between the published deletions carried by JBS patients with immunodeficiency [5,8,9,10,11] (Bernaciak et al., 2008 [8]; Blazina et al., 2016 [9], Grossfels et al., 2004 [10], Penny et al., 1995 [11]) and the deletion carried by our patient. On the right, the deleted genes in our patient are detailed. Haploinsufficiency of the *FLI-1* gene has been proposed as the genetic alteration responsible for the immune system defects in JBS, for both our patient and other patients he has been compared with.

**Table 1 genes-12-01197-t001:** Deleted OMIM genes in the 11q24.3q25 deletion in our JBS patient.

Gene Symbol	OMIM Number	OMIM Gene Title	Start–End Gene Position in Chromosome 11	Size (kbp)	OMIM Associated Disorder
*ETS1*	164720	Avian erythroblastosis virus E26 (v-ets) oncogene homolog-1	128,328,655–128,457,453	129	
*FLI1*	193067	Friend leukemia virus integration 1	128,556,429–128,683,162	127	
*KCNJ1*	600359	Potassium inwardly-rectifying channel, subfamily J, member 1	128,707,908–128,737,268	29	Bartter syndrome, type 2, 241200
*KCNJ5*	600734	Potassium inwardly-rectifying channel, subfamily J, member 5	128,761,312–128,787,951	27	Long QT syndrome 13, 613485; hyperaldosteronism, familial, type III, 613677
*TP53AIP1*	605426	p53-regulated apoptosis-inducing protein-1	128,804,626–128,813,294	9	
*ARHGAP32*	608541	Rho GTPase activating protein 32	128,834,954–129,062,093	227	
*BARX2*	604823	BarH-like homeo box gene 2	129,245,880–129,322,174	76	
*NFRKB*	164013	Nuclear factor related to kappa B-binding protein	129,733,669–129,765,490	32	
*APLP2*	104776	Amyloid beta (A4) precursor-like protein-2	129,939,715–130,014,706	75	
*ST14*	606797	Suppression of tumorigenicity 14	130,029,681–130,080,257	51	Ichthyosis with hypotrichosis, 610765
*ADAMTS8*	605175	A disintegrin-like and metalloproteinase with thrombospondin type-1motif, 8	130,274,817–130,298,539	24	
*ADAMTS15*	607509	A disintegrin-like and metalloproteinase with thrombospondin type-1motif, 15	130,318,868–130,346,539	28	
*NTM*	607938	Neurotrimin	131,240,370–132,206,716	966	
*OPCML*	600632	Opioid-binding protein/cell adhesion molecule-like	132,284,874–133,402,403	1,118	Ovarian cancer, somatic, 167000
*SPATA19*	609805	Spermatogenesis-associated protein 19	133,710,516–133,715,392	5	
*IGSF9B*	613773	Immunoglobulin superfamily, member 9B	133,778,519–133,826,649	48	
*JAM3*	606871	Junctional adhesion molecule 3	133,938,819–134,021,652	83	Hemorrhagic destruction of the brain, subependymal calcification, cataracts, 613730
*NCAPD3*	609276	Non-SMC condensin II complex subunit D3	134,022,336–134,094,426	72	
*VPS26B*	610027	Vacuolar protein sorting 26, yeast, homolog of, B	134,094,560–134,117,686	23	
*THYN1*	613739	Thymocyte nuclear protein 1	134,118,172–134,123,260	5	
*ACAD8*	604773	Acyl-CoA dehydrogenase family, member 8	134,123,433–134,135,746	12	Isobutyryl-CoA dehydrogenase deficiency, 611283
*B3GT1*	151290	Beta-1,3-glucuronyltransferase 1	134,248,397–134,281,812	33	

The table presents the gene symbol, the OMIM number, the OMIM gene title, the position in chromosome 11, the size and the OMIM associated disorders possible for the deleted genes of our JBS patient.

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
