# Peer review of "Immune Deficiency in Jacobsen Syndrome: Molecular and Phenotypic Characterization"

_genes, 2021, doi:10.3390/genes12081197_

Round 1

Reviewer 1 Report

The manuscript is a case report describing Jacobsen Syndrome caused by the deletion in the terminal part of  11q. The manuscript is interesting and nicely designed. It shows unusual case important for future doctor's assessment. Only minor comments for the paper are considered as the following:

  1. Do we have an agreement for publicizing the child's faces? Page 3, Figure 1
  2. For the set of deleted genes in 11q24.3-q25, perform in silico enrichment analysis of these genes to investigate the functions of genes in clusters. Although the description of genes in the Discussion section is already nicely described, however an enrichment analysis can improve the description.

Author Response

The manuscript is a case report describing Jacobsen Syndrome caused by the deletion in the terminal part of  11q. The manuscript is interesting and nicely designed. It shows unusual case important for future doctor's assessment.

Thank you very much for rating our manuscript as interesting and well designed, thank you for taking our efforts into account. Indeed, our interest has been mainly to provide work methodology for clinicians who attend the difficult neuropediatric consultations.

Only minor comments for the paper are considered as the following:

Point 1. Do we have an agreement to publicize the child's faces? Page 3, Figure 1.

No. The parents have been extraordinarily helpful and gracious at every visit. We ourselves were the ones who advised to avoid publishing the patient's face, to avoid any psychological damage that could be caused by being seen on the internet. We currently consider that the clinical benefit it provides is not justified due to the psychological impact it could have, even more associated with the text we publish, on a young man with the characteristics of this patient.

Point 2. For the set of genes deleted at 11q24.3-q25, perform an in silico enrichment analysis of these genes to investigate the functions of the genes in clusters. Although the description of genes in the Discussion section is already well described, however, an enrichment analysis can improve the description.

Indeed, these days we have been studying the existing tools for what you propose, and they are spectacular: The Gene Ontology (GO) and Kyoto Encyclopedia of Gene and Genome (KEGG) pathway analysis, using the database for annotation, visualization and integration. Discovery (DAVID). Also the protein-protein interaction network (PPI), mediated by the search tool for the recovery of interacting genes (STRING). We have spent time trying to contribute information to the manuscript, but we need more time to master and control these resources. Sorry, we're not used to it. We have not achieved it in the days that the editors give us to answer. It's a great suggestion. We use the Human Phenotype Ontology and Face2Gene, as the main tools for "phenotyping", but of course we will work to incorporate these resources with infinite possibilities.

Reviewer 2 Report

Nice report

Author Response

We believe that reviewer number 2 has not made any suggestions. Thanks for your effort.

Reviewer 3 Report

Jacobsen syndrome or JBS also known as 11q terminal deletion syndrome, is a rare genetic disorder caused by the loss of a continuous set of 37 genes located on the long arm of chromosome 11. The size of deletion 38 ranges from 2.9 to 20 Many studies have made it possible to clarify aspects both from a clinical point of view and from a diagnosis of an orphan genetic pathology.
As this pathology is complex it certainly cannot be entrusted to a software for the final diagnosis, but only starting with a correct differential diagnosis.
The somatic traits not fully highlighted by the HPO database and Face2Gene software could be explained by a possible mosaicism.
How has the exclusion of mosaicism you declared been demonstrated? See line 166.
Did parental analysis use only classical karyotype? Given the enormous court of analysis that you have submitted to the child, a comparative analysis could also be extended to parents the whole genome SNPs / CNVs microarray hybridization. See line 159-161.

Author Response

Jacobsen syndrome or JBS, also known as terminal 11q deletion syndrome, is a rare genetic disorder caused by the loss of a continuous set of 37 genes located on the long arm of chromosome 11. The size of the 38 deletion varies from 2.9 to 20. allowed to clarify aspects both from the clinical point of view and from the diagnosis of an orphan genetic pathology.

As this pathology is complex, it certainly cannot be entrusted to software for the final diagnosis, but rather from a correct differential diagnosis.

Point 1. Somatic features not fully highlighted by the HPO database and Face2Gene software could be explained by possible mosaicism. How has the exclusion of the mosaicism you stated been demonstrated? See line 166.

Thank you for raising this possibility, it could certainly be a response to the very differential phenotype. We have ruled out that the patient carries the mosaic line deletion, because we performed their analysis using the high-resolution whole genome SNPs / CNVs microarray hybridization Cytogenetics CytoScan® HD Array (Affymetrix Inc., Santa Clara, CA, USA). We have been using the different versions of this array since 2007, and its design with a large number of probes and markers (more than 3 million markers covering the entire genome, in CytoScan HD) ALLOWS TO DETECT MOSAIC LINES in a very low percentage. In the case of our patient, the fluorescence signals left no doubt that the alteration was in full line and heterozygous.

Point 2. Did the analysis of the parents use only the classical karyotype? Given the huge court of analysis that the child has presented, a comparative analysis could also be extended to parents on genome-wide SNP / CNV microarray hybridization. See line 159-161.

Indeed, we conducted the study in both parents only with a conventional karyotype. Both karyotypes were absolutely normal, and both ruled out with complete certainty the presence of the deletion of the patient, which, due to its large size (6.8 Mb), is perfectly visible with this conventional technique.

However, again the reviewer raises a great suggestion. It made us think that it would have been great to have carried out the study of the parents with the same high-density array technology that was used in the patient. The justification is NOT to expect them to carry a minor deletion (and hence the lack of phenotype in the parents) but, as the reviewer comments, the possibility of comparing the LOH regions, as well as CNVs carried by the patient, father and mother. The studies that we have carried out in this way in other cases of syndromic patients reveal that there is a lot to investigate in the heritability of the LOH, CNVs and SVs regions, as well as the importance of the coincidence of the LOH regions in parents of patients with cytogenetic alterations.

We have no possibility to do it now since, as we have commented to editors and other reviewers, we have made great efforts to reconnect with the family and we have not been able to do so; we consider that they have left our country. No DNA sample was extracted from the parents, but the specific one to perform a conventional karyotype; We will take into account the reviewer's consideration to carry out in our cases from now on.